# Acclimatization of In Vitro Banana Seedlings Using Root-Applied Bio-Nanofertilizer of Copper and Selenium

Tarek A. Shalaby [1,2,*], Said M. El-Bialy [2], Mohammed E. El-Mahrouk [2], Alaa El-Dein Omara [3], Hossam S. El-Beltagi [4,5,*] and Hassan El-Ramady [6,7]

1 Department of Arid Land Agriculture, College of Agricultural and Food Science, King Faisal University, Al-Ahsa 31982, Saudi Arabia

2 Horticulture Department, Faculty of Agriculture, Kafrelsheikh University, Kafr El-Sheikh 33516, Egypt; saeedelbiely@gmail.com (S.M.E.-B.); threemelmahrouk@yahoo.com (M.E.E.-M.)

3 Agriculture Microbiology Department, Soil, Water and Environment Research Institute (SWERI), Sakha Agricultural Research Station, Agriculture Research Center (ARC), Kafr El-Sheikh 33717, Egypt; alaa.omara@yahoo.com

4 Agricultural Biotechnology Department, College of Agriculture and Food Sciences, King Faisal University, Al-Ahsa 31982, Saudi Arabia

5 Biochemistry Department, Faculty of Agriculture, Cairo University, Giza 12613, Egypt

6 Soil and Water Department, Faculty of Agriculture, Kafrelsheikh University, Kafr El-Sheikh 33516, Egypt; hassan.elramady@agr.kfs.edu.eg

7 Institute of Animal Science, Biotechnology and Nature Conservation, Faculty of Agricultural and Food Sciences and Environmental Management, University of Debrecen, 138 Böszörményi Street, 4032 Debrecen, Hungary

* Correspondence: tshalaby@kfu.edu.sa (T.A.S.); helbeltagi@kfu.edu.sa (H.S.E.-B.)

**Abstract:** The production of in vitro banana transplants has become an important practice in the global banana production. Proper and enough nutrients are needed for banana production particularly during the acclimatization period. To avoid the environmental problem resulting from the chemical fertilizers, nanofertilizers of Se and Cu were separately applied during the acclimatization of banana. The biological form of nano-Cu (50 and 100 mg $L^{-1}$) and nano-Se (25, 50, 75, and 100 mg $L^{-1}$) were studied on acclimatized banana transplants under greenhouse conditions. Both applied nanofertilizers enhanced the growth of transplant by 10.9 and 12.6% for dry weight after nano-Se and nano-Cu application up to 100 mg $L^{-1}$, respectively. The survival rate was also increased by increasing applied doses of both nanofertilizers up to 100 mg $L^{-1}$, whereas the highest survival rate (95.3%) was recorded for nano-Cu. All studied photosynthetic pigments and its fluorescence were improved by applying nanofertilizers. Studied antioxidant enzymatic activities (CAT, PPO, and POX) were also increased. A pH decrease in the growing medium was noticed after applying nano-Cu, which may explain the high bioavailability of studied nutrients (N, P, K, Cu, Fe, Se, and Zn) by banana transplants.

**Keywords:** *Musa* spp.; catalase; polyphenol oxidase; peroxidase; chlorophyll; photosynthetic pigments

## 1. Introduction

Several environmental problems have been observed from the intensive use of conventional mineral fertilizers, which induced the pollution of foods and the degradation of soils [1]. These mineral conventional fertilizers have a low use efficiency (ranging from 20–40%) and a high leaching rate of nutrients, such as N and P, to groundwater and rivers, which cause the eutrophication phenomena and many problems for human health [2]. Nanofertilizers have the properties of both nanoparticles and nutrients themselves, so they are considered promising fertilizers [3]. The nanofertilizers are materials with the ability to enhance plant grow and its productivity in a better manner compared to chemical fertilizers, which include nano-zeolite, macro-nutrient nanofertilizers, nano-hydroxyapatite, and nano-biofertilizers [4]. Many nutrients have been applied as nutrient-based nanofertilizers such



as iron [5], silicon [6], zinc [7], NPK [8], copper [9], and selenium [10]. Several studies have been conducted on the benefits of nanofertilizers on crops such as cabbage [5], maize [6], and wheat [7], but there is currently no published paper on the banana crop, although banana peels have been used in producing nanofertilizers [8]. Concerning the benefits of nanofertilizers, Sheoran et al. [7] confirmed improving fruit quality, productivity, shelf life, and reducing the leaching of nutrients in soil after the harvesting of crops.

Banana (*Musa* spp.) is significant fruit crop cultivated in the tropical and sub-tropical regions [11] with a high yield production (up to 35 Mg ha$^{-1}$) and high requirement of NPK-fertilizers, ranging from 0.19–0.34, 0.022–0.146, and 0.73–1.26 Mg ha$^{-1}$[12–14]. Banana is classified economically as the fifth most important agricultural crop in global trade [13]. Banana fruits have excellent nutritional value due to their high content of carbohydrates, minerals (K, P, Ca, Mg, and Fe), and vitamins like riboflavin (B2), niacin (B3), A, C and B6 [14,15]. The growth, rooting and acclimatization of micropropagated banana transplants may be controlled by many factors like endophytic micro-organisms [16], aseptic environments [17], nutrients such as magnesium [18], potassium, and nitrogen [19], and growth media content [20]. Micropropagated banana transplants should have high health and be cultivated in suitable artificial culture media, free from diseases and pests to guarantee a high production [17]. The acclimatization of in vitro grown banana plantlets is a very important step for commercial production in an *ex-vitro* environment by hardening for high quality banana transplants as followed for several horticultural crops like orchid [21]. As far as we know, this is the first report about the rooting and acclimatization of microprogrammed banana under the fertilization of Se and/or Cu bio-nanofertilizers.

Therefore, this study was an attempt to evaluate the soil application of Se and Cu bio-nanofertilizers on the rooting and acclimatization of in vitro-grown banana plantlets. More open questions are subjected to be answered in the current study such as: what are the expected economical and botanical benefits of applied bio-nanofertilizers on banana transplants during acclimatization? What are the subjected gains from these bio-nanofertilizers on banana productivity in the short- and long-term?

## 2. Materials and Methods

### 2.1. Plant Materials and Growing Media

In vitro shoots of Musa sp. 'Grand nain' were rooted on MS medium [22] semi-solid medium (7.0 g L$^{-1}$ ager) enhanced with 30 g L$^{-1}$ sucrose, 1 mg L$^{-1}$ Indole butyric acid ((IBA). Well-developed plantlets (3–5 leaves, 7–8 cm in height and 2–3 g in weight) were used in this study, which were bought from Kinana Company, Kafr El-Zayat, Gharbia Governorate. Acclimatized growing medium was prepared by mixture of 300 L peatmoss + 1 kg foam + 30 kg vermic-ulite. The experimental unit was plastic cup (5 cm). The growing medium was sterilized with a fungicidal solution (1 g L$^{-1}$ Rizolex; Kafr El-Zayat Company, El-Gharbia, Egypt). The pH of used medium was adjusted to 6 $\pm$ 1 with calcium carbonate powder using a pH meter (Jenway 3510, Staffordshire, UK). Each plastic cup contained only one plantlet and the number of replicates was 10 plantlets.

### 2.2. Preparing of Bio-Nanofertilizers

Biological selenium and copper nanoparticles were produced at the Agricultural Microbiology Lab, Soil, Water and Environment Research Institute, Agricultural Research Center, Giza, Egypt. High resolution transmission electron microscope (HR-TEM, Tecnai G20, FEI, Amsterdam, The Netherlands) was used for measuring the dimension of selenium and copper nanoparticles (100–300 and 350–500 nm, respectively), which were biosynthesized and analyzed by Nanotechnology and Advanced Material Central Lab, Agriculture Research Center (ARC), Egypt.

### 2.3. Acclimatization Treatments

The experiment was carried out in Tahrir Directorate, Badr Center (30°08′9.60″ N 31°42′54.00″ E), El Bahira Gover-norate, Egypt, in a private nursery using agricultural

greenhouse (9 × 40 m). Plantlets or transplants were cultivated on 17 October 2018 for the acclimatization and rooting of transplants of banana, which continued for 45 days. On 17 October 2018 transplants also received the first dose with different concentrations of nano-selenium (25, 50, 75 and 100 mg $L^{-1}$) and nano-copper (50, 100 mg $L^{-1}$) as a soil application. After 30 days, the second doses of nano-selenium and copper were applied. Transplants were grown under 50% shade net and a compound fertilizer solution (N:P:K at 19:19:19) as water-soluble fertilizer (Rosasol; Rosier, Moustier, Belgium) at 1 g $L^{-1}$ was used once after 35 days from culture in the irrigation water. Some vegetative parameters were measured to evaluate the rooting and acclimatization of banana including transplant height, number of leaves, transplant diameter, root number and its length, transplant dry weight and survival percent, which was recorded after 45 days. This period represents the maximum period for acclimatization of banana in horticultural nurseries.

### 2.4. Measuring Photosynthetic Parameters

#### 2.4.1. Chlorophyll Pigments

Spectrophotometric analysis was performed to examine the quantity of chlorophyll a (Chl a), chlorophyll b (Chl b), and carotenoids in fully expanded young leaves. Chlorophyll was extracted from leaf tissue by immersing 1.0 g fresh leaf in 5 mL N,N-Dimethyl formamide for 48 h at 4 °C in dark. The absorbance of Chl a, -b, and carotenoids was evaluated using a spectrophotometer (Double beam UV/Visible Spectrophotometer Libra S80PC, Cambridge England) at 665, 649, and 470 nm, respectively. The spectrophotometric data were used to calculate chlorophyll and carotenoid concentrations using the Moran and Porath formula [23]. Each treatment was replicated three times.

#### 2.4.2. Chlorophyll Fluorescence

Chlorophyll fluorescence parameters were measured on the abaxial surface of freshly detached leaf discs, which included keeping plants for 30 min in the dark prior to measurement. A handheld chlorophyll fluorescence meter was used to measure modulated fluorescence (OS30P, Labo Amirica, Fremont, CA, USA). Minimal fluorescence (F0) was evaluated in dark-adapted leaves for 30 min utilizing light of 0.1 mol $m^{-2}$ $s^{-1}$, and maximal fluorescence (Fm) was tested in the same leaves after one second (s) saturating pulse (>3500 mol $m^{-2}$ $s^{-1}$). According to Dewir et al. [24], the maximum variable fluorescence (Fv = Fm − F0) and photo-chemical efficiency of PSII (Fv/Fm) were calculated for dark-adapted leaves utilizing four randomly selected plants and a standard leaf chamber on fully expanded young leaves. Within each treatment, there were four single-leaf replications.

### 2.5. Biochemical Assessments

A total of 0.5 g of fully expanded leaves were homogenized in liquid nitrogen with 3 mL of extraction buffer 50 mM TRIS buffer (pH 7.8), adding 1 mM EDTA-$Na_2$ and 7.5% polyvinylpyrrolidone to measure antioxidant enzymatic activities. Using a previously chilled mortar and pestle, the homogenate was centrifuged at 12,000 rpm for 20 min at 4 °C after being filtered through four layers of cheesecloth. According to Hafez et al. [25], supernatant was re-centrifuged at 12,000 rpm for 20 min at 4 °C to determine total soluble enzyme activity utilizing UV-spectrophotometer (160 A—Shimadzu, Japan). Catalase (CAT), polyphenol oxidase (PPO), and peroxidase (POX) activities were determined [26–28] using three replicates.

### 2.6. Analyses of Growing Media

The analyses of growing medium were performed before the experiment including pH in a 1:1 ratio using a calibrated pH meter (Jenway 3510, Staffordshire, Chelmsford Essex, UK) and salinity or electrical conductivity (EC) in 1:5 ratio utilizing EC Meter (MI 170, Milan, Italy). Available phosphorus was measured according to Olsen and Sommers [29], whereas available potassium was determined by flame photometer (Jenway PFP7, Staffordshire,

UK). The available copper, iron, manganese, and zinc as well as selenium were quantified using an atomic absorption spectrophotometer (Avanta E, GBC, Victoria, Australia) [30].

### 2.7. Chemical Composition of Acclimatized Plants

Plant samples were oven-dried at 65 °C for 48 h to determine dry weights before being ground into homogeneous powder in metal-free mill (IKa-Werke, M 20 Darmstadt, Germany). Phosphorus and potassium were measured using spectrophotometer (GT 80+, Livingston, UK) and atomic absorption spectrometry method [31,32], respectively. The concentrations of Cu, Fe, Mn, and Zn were measured by atomic absorption spectrometry (Avanta E; GBC) [30]. The instrument of hydride generation atomic fluorescence spectroscopy was used in measuring the Se content in growing media and plant samples according to Dernovics et al. [33]. All previous parameters were measured using three replicates.

### 2.8. Statistical Analyses

Experiments were established in highly randomized design, with three replicates for each treatment. Ten transplants were used to reflect each replicate. Observations on acclimatized plants were recorded after 45 days of culture. SPSS software was used to perform a variance analysis on the data (version 20; IBM Corp., Armonk, NY, USA). Duncan's multiple range testing method was used to perform the mean separations, and significance was determined at $p \leq 0.05$.

## 3. Results

### 3.1. Vegetative Growth of Acclimatized Plants

Before cultivating the banana transplants, the pH of the growing medium and its salinity (EC) were measured to be 5.0 and 0.53 dS m$^{-1}$, respectively. After harvesting, both the pH and EC of the growing medium were measured again as presented in Table 1. The values in pH of the growing medium increased significantly with increasing nano-Se doses till 75 mg L$^{-1}$, whereas these values were decreased with increasing doses of nano-Cu up to 100 mg L$^{-1}$ (Table 1). The highest value in EC (0.479 dS m$^{-1}$) was recorded for nano-Se at a dose of 75 mg L$^{-1}$, whereas the values of EC were decreased with increasing application of nano-Cu. During the acclimatization, some vegetative growth parameters were measured for banana transplants (Figures 1 and 2). In general, all studied vegetative growth parameters were higher at the highest applied doses of nano-Se and nano-Cu, with priority to nano-Cu, which recorded the highest values in all studied vegetative parameters at the dose of 100 mg kg$^{-1}$ compared to values in the case of nano-Se (Figure 3). The most important parameter for the acclimatized plantlets is the survival rate (%), which represents extent in which the applied nanofertilizers can support the acclimatization of banana transplants. The acclimatized banana transplants recorded the highest survival rate up to 95.3% for an applied 100 mg kg$^{-1}$ nano-Cu and increased with increased applied doses of both nano-Se and nano-Cu.

**Table 1.** The pH and salinity (EC) of growing medium after harvesting the banana transplants under different concentrations of nano-Se and nano-Cu.

| Treatments (mg L$^{-1}$) | Growing Medium pH | Salinity of Growing Medium (dS m$^{-1}$) |
|---|---|---|
| T1: Control | 5.42 | 0.435 |
| T2: Nano-Se (25) | 5.32 | 0.446 |
| T3: Nano-Se (50) | 5.41 | 0.479 |
| T4: Nano-Se (75) | 5.77 | 0.330 |
| T5: Nano-Se (100) | 5.52 | 0.329 |
| T6: Nano-Cu (50) | 5.27 | 0.413 |
| T7: Nano-Cu (100) | 5.25 | 0.410 |

Note: growing medium pH and its salinity (EC) were 5.0 and 0.53 dS m$^{-1}$, respectively, before cultivation.

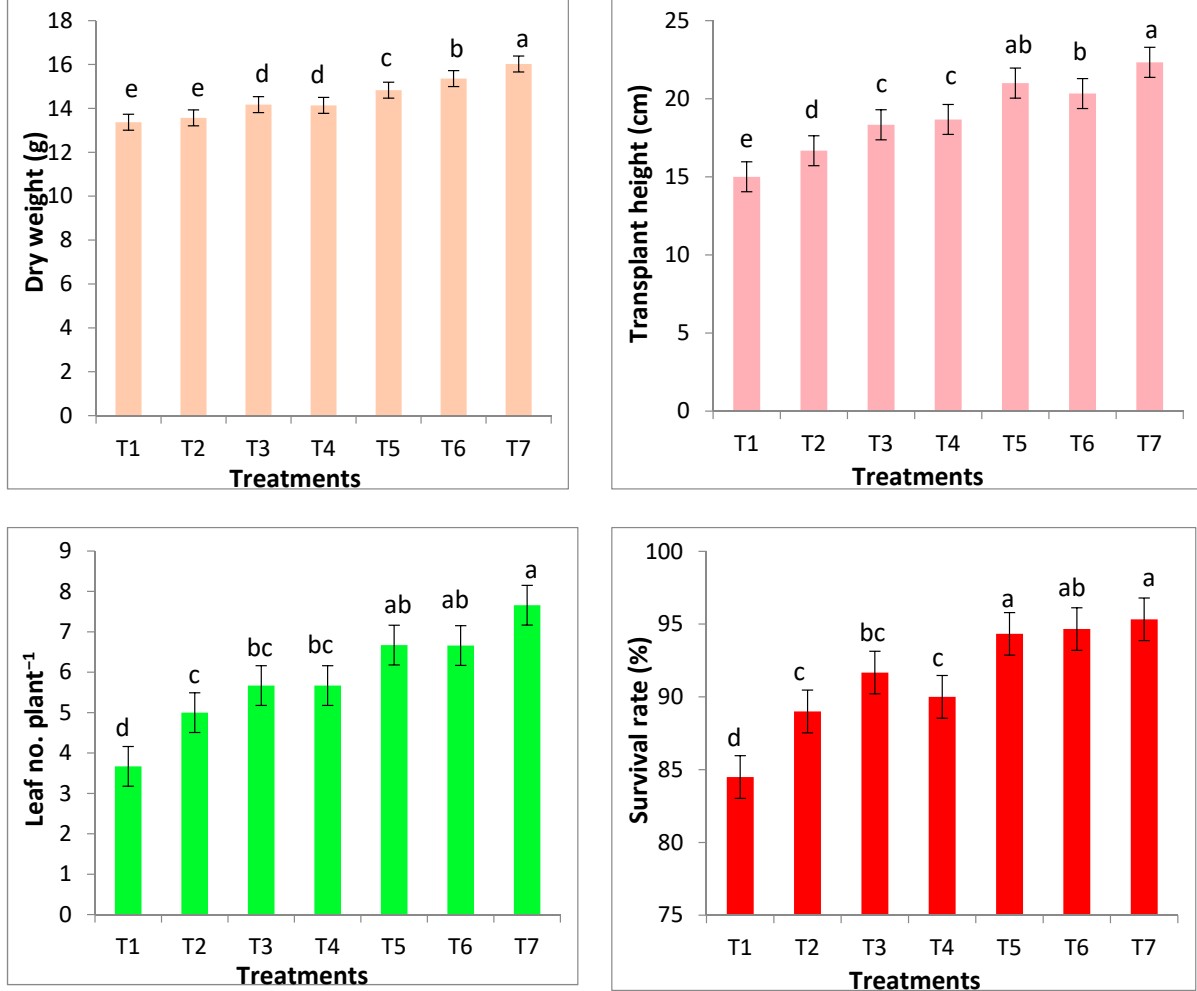

**Figure 1.** Effect of different concentrations of nano-Se and nano-Cu on some vegetative growth of banana transplants (dry weight, height, leaves number per plant), survival rate %. (Abbreviations: T1, T2, T3, T4 and T5 represent control, 25, 50, 75, 100 mg L$^{-1}$ nano-Se, respectively, whereas T6 and T7 represent 50, 100 mg L$^{-1}$ nano-Cu. Values are means ± standard deviation (SD) from three replicates.) Columns have the same letter are not significant according to Duncan's multiple range test at $p \leq 0.05$.

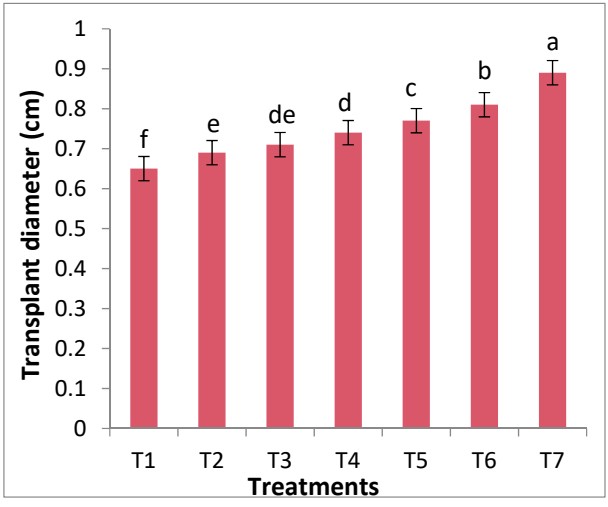

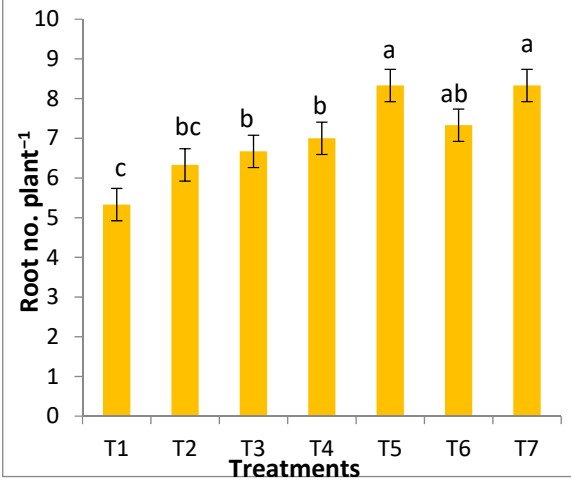

**Figure 2.** *Cont.*

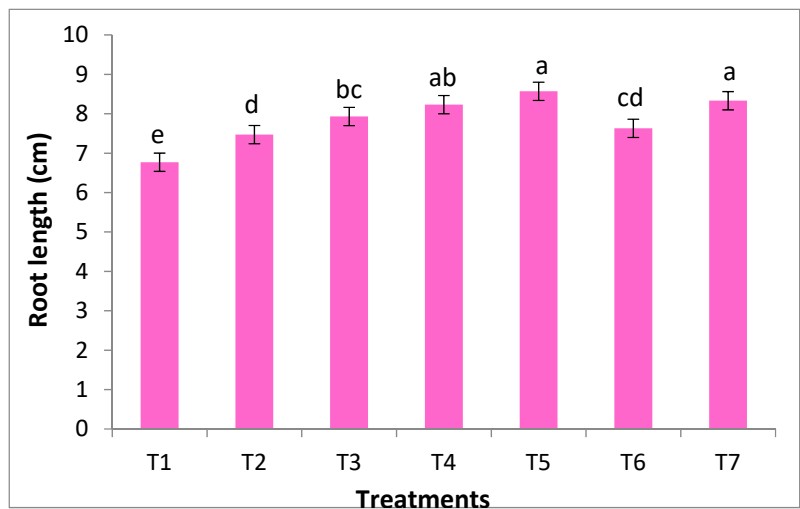

**Figure 2.** Effect of different concentrations of nano-Se and nano-Cu on some vegetative growth of banana plants (plant diameter, root height per plant and number of roots per plant). (Abbreviations: T1, T2, T3, T4 and T5 represent control, 25, 50, 75, 100 mg L$^{-1}$ nano- Se, respectively, whereas T6 and T7 represent 50, 100 mg L$^{-1}$ nano-Cu. Values are means ± standard deviation (SD) from three replicates.) Columns have the same letter are not significant according to Duncan's multiple range test at $p \leq 0.05$.

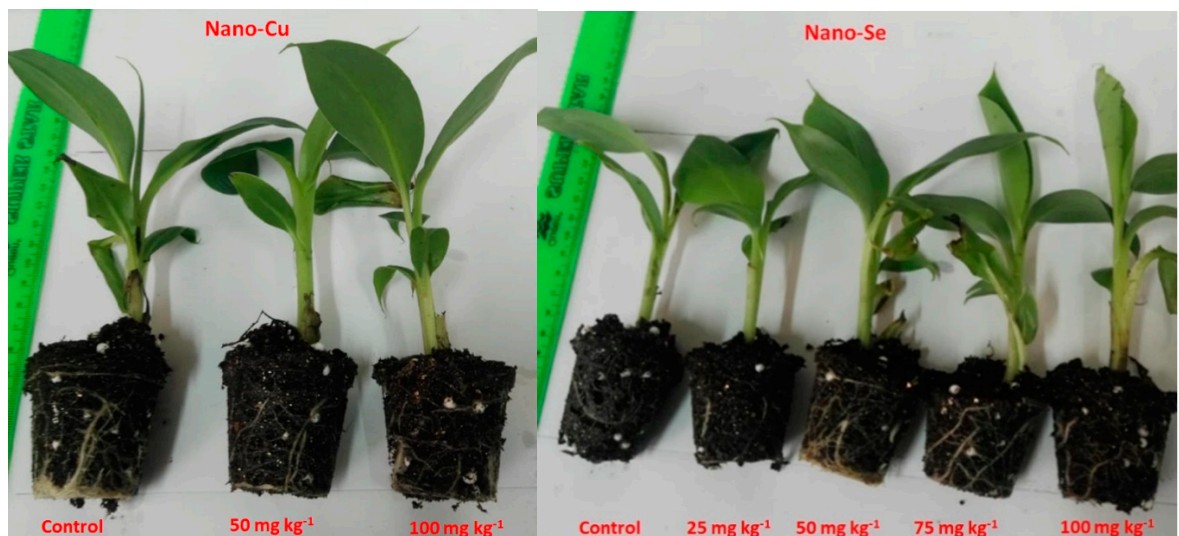

**Figure 3.** General photos for different treatments, which present different doses of applied nano-Se and nano-Cu.

### 3.2. Photosynthetic Pigments and Its Fluorescence

Based on the importance of photosynthesis in banana transplants and its efficiency, the photosynthetic parameters in the current study were measured (Figure 4). Total chlorophyll (i.e., a and b) and carotenoids were increased significantly by increasing the applied doses of nano-Se and the same response was found for nano-Cu. Highest values of total chlorophyll and carotenoids (2.21 and 0.56) mg g$^{-1}$ FW were obtained under the highest applied dose of nano-Se on banana transplants.

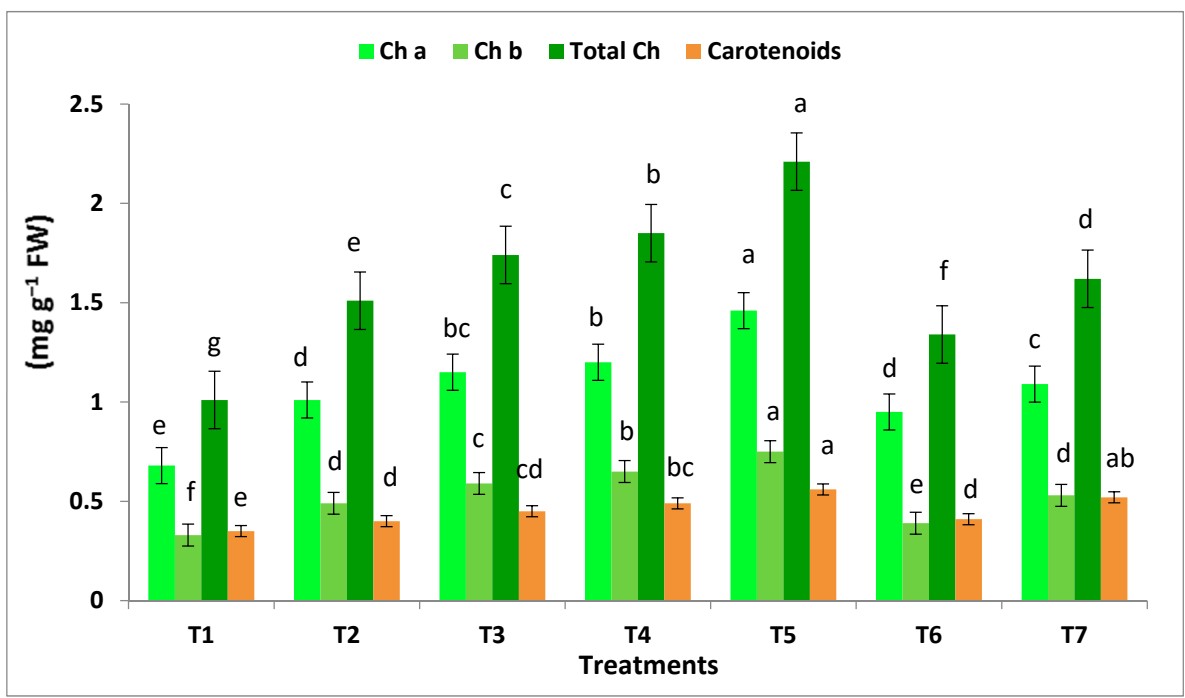

**Figure 4.** Effect of different concentrations of nano-Se particles on Ch a, Ch b, total Ch and carotenoids in banana leaves. (Abbreviations: T1, T2, T3, T4 and T5 represent control, 25, 50, 75, 100 mg L$^{-1}$ nano-Se, respectively, whereas T6 and T7 represent 50, 100 mg L$^{-1}$ nano-Cu. Values are means $\pm$ standard deviation (SD) from three replicates). Columns have the same letter are not significant according to Duncan's multiple range test at $p \leq 0.05$.

Banana plants' photosynthetic fluorescence parameters, including minimum fluorescence (F0), highest fluorescence (Fm), range of fluorescence (Fv = Fm − F0), were evaluated. The values of the various parameters were increased significantly by increasing applied doses of both nano-Se and nano-Cu, whereas the values of photochemical efficiency of PSII ($F_V/F_0$), and highest quantum efficiency of photosystem II (Fv/Fm) have the opposite trend. Under nano-Cu fertilization, the values of $F_0$, $F_m$, and the variable fluorescence (502, 1694, and 1192, respectively) were the highest compared to values of nano-Se (Table 2).

**Table 2.** Impact of various nano-Se and nano-Cu concentrations on different photosynthetic variables in banana plants such as FO, FV, FM, FV/FM, and FV/F0.

| Treatments (mg L$^{-1}$) | F0 | FM | FV | FV/FM | FV/F0 |
|---|---|---|---|---|---|
| T1: Control | 375 $\pm$ 4.73 f | 1517 $\pm$ 8.08 f | 1142 $\pm$ 4.58 b | 0.75 $\pm$ 0.02 a | 3.04 $\pm$ 0.07 a |
| T2: Nano-Se (25) | 396 $\pm$ 7.21 e | 1541 $\pm$ 3.06 e | 1145 $\pm$ 8.50 b | 0.74 $\pm$ 0.02 b | 2.89 $\pm$ 0.04 b |
| T3: Nano-Se (50) | 422 $\pm$ 7.37 d | 1569 $\pm$ 4.04 d | 1147 $\pm$ 7.21 b | 0.73 $\pm$ 0.01 c | 2.71 $\pm$ 0.04 c |
| T4: Nano-Se (75) | 460 $\pm$ 6.11 c | 1592 $\pm$ 4.36 c | 1132 $\pm$ 9.00 c | 0.71 $\pm$ 0.01 d | 2.46 $\pm$ 0.04 d |
| T5: Nano-Se (100) | 474 $\pm$ 4.16 b | 1600 $\pm$ 5.51 c | 1126 $\pm$ 6.66 d | 0.70 $\pm$ 0.01 d | 2.37 $\pm$ 0.05 d |
| T6: Nano-Cu (50) | 451 $\pm$ 8.00 c | 1644 $\pm$ 8.19 b | 1193 $\pm$ 3.66 a | 0.72 $\pm$ 0.02 c | 2.65 $\pm$ 0.04 c |
| T7: Nano-Cu (100) | 502 $\pm$ 10.79 a | 1694 $\pm$ 7.09 a | 1192 $\pm$ 2.21 a | 0.70 $\pm$ 0.04 d | 2.37 $\pm$ 0.06 d |
| L.S.D 0.05 | 12.62 | 10.62 | 5.40 | 0.01 | 0.08 |

Means in column followed by common letter are not significantly different at the 5% level by DMRT; values are means $\pm$ standard deviation (SD) from three replicates. Minimum fluorescence (F0), maximum fluorescence (Fm), variable fluorescence (Fv = Fm − F0), photochemical efficiency of PSII (FV/F0), maximum quantum efficiency of photosystem II (Fv/Fm).

### 3.3. Enzymatic Antioxidant Activities

The response of banana transplants to different applied doses of nano-Se or nano-Cu during acclimatization is represented in increasing contents of enzymatic antioxidants

including catalase, peroxidase, and polyphenol oxidase (Figure 5). It is clear that, all previous enzymatic antioxidants were significantly increased by increasing the applied doses of nano-Se and nano-Cu. The highest values of CAT, POX, and PPO enzyme activities were achieved when the highest applied doses of nano-Se and nano-Cu were added to growing media. This may clarify why these nanofertilizers can support transplants of banana during their acclimatization.

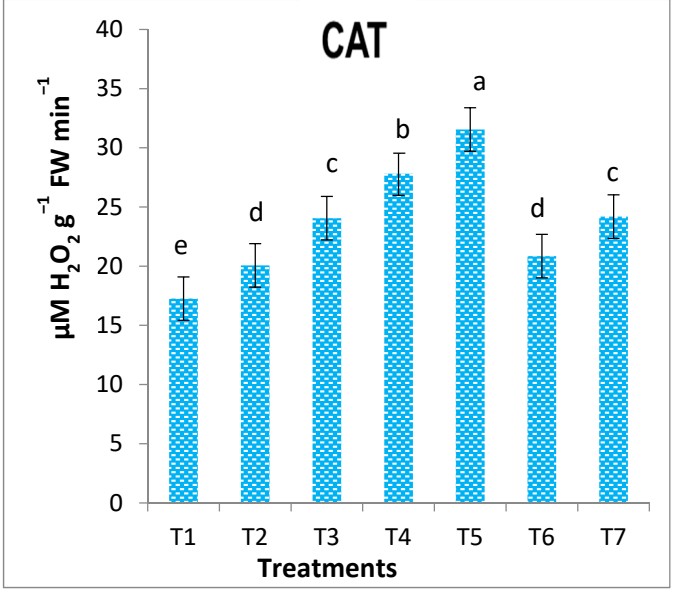
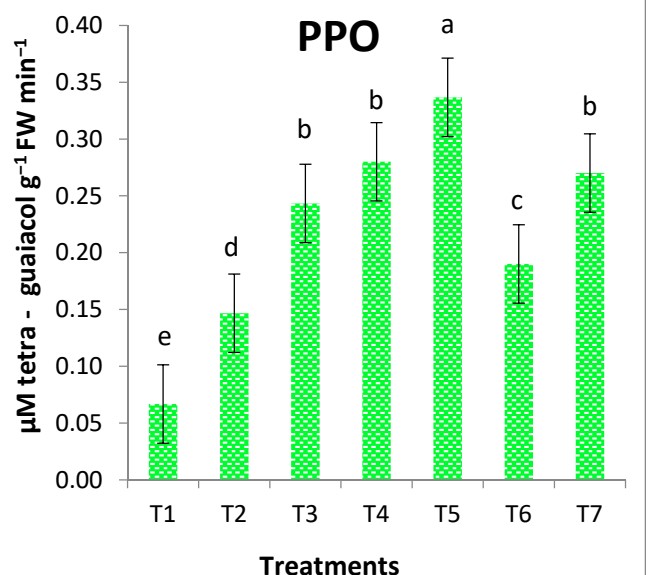

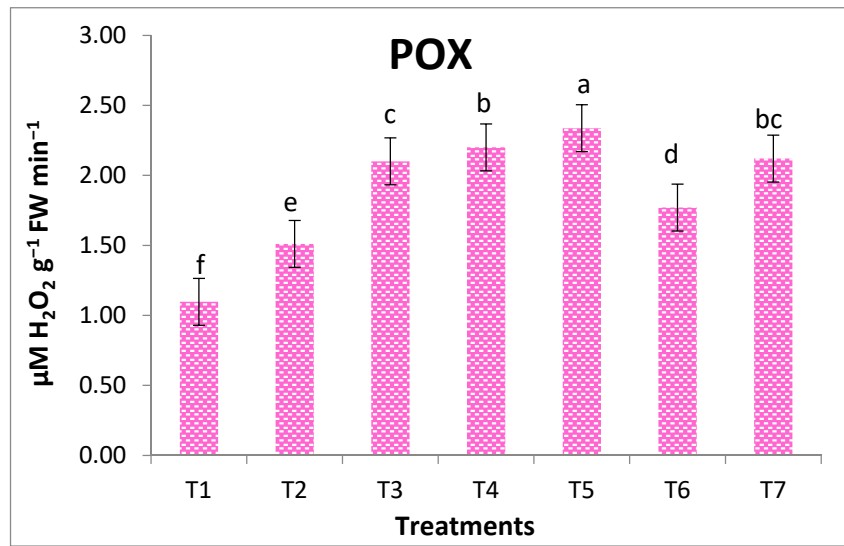

**Figure 5.** Effect of different concentrations of nano-Se particles on antioxidant activities, catalase (CAT), poly phenol oxidase (PPO) and peroxidase (POX) in banana leaves. (Abbreviations: T1, T2, T3, T4 and T5 represent control, 25, 50, 75, 100 mg L$^{-1}$ nano-Se, respectively, whereas T6 and T7 represent 50, 100 mg L$^{-1}$ nano-Cu. Values are means $\pm$ standard deviation (SD) from three replicates.) Columns have the same letter are not significant according to Duncan's multiple range test at $p \leq 0.05$.

### 3.4. Applied Nanofertilizers and Growing Medium

To know what does happen after the application of nanofertilizers to used growing medium, soil samples were gathered from growing cups and analyzed. Some nutrients, such as N, P, K, Fe, Mn, Zn, Cu, Se, were measured in growing media (Table 3). It could be noticed that nano-Se increased the content of Se in measured treatments with increasing

applied doses of nano-Se, as reported in the case of nano-Cu, which had the same trend. Thus, the highest measured Se content (0.0057 mg kg$^{-1}$) resulted from the application of 100 mg L$^{-1}$ nano-Se compared to the other treatments. The same trend was observed for the application of 100 nano-Cu, which recorded the highest concentration in growing media in all measured treatments (141.8 mg kg$^{-1}$) as reported in Table 3. As distinguished in Table 4, the applied nano-Se (up to 100 mg L$^{-1}$) decreased the soil content of P due to an antagonism effect, but there is not a clear trend for K, with some divergent changes between nano-Se rates. This trend also was observed in the case of applied nano-Cu, for which the highest applied dose (100 mg L$^{-1}$) recorded a decrease in N, Fe, and Mn values compared to the lower applied dose (50 mg L$^{-1}$) and control. This may reflect the antagonism effect, which resulted from applied high doses of nano-Cu on the bioavailability of other nutrients in soils.

**Table 3.** Effect of applied nano-Cu and nano-Se on bioavailability of some nutrient contents (mg kg$^{-1}$) in growing medium.

| Doses | N | P | K | Fe | Mn | Zn | Cu | Se |
|---|---|---|---|---|---|---|---|---|
| T1: Control | 112 ± 1.00 g | 14.60 ± 0.10 b | 191.56 ± 2.95 c | 232 ± 1.55 d | 17.5 ± 0.50 c | 28.00 ± 2.00 ab | 6.77 ± 0.25 d | 0.0015 ± 0.0001 e |
| T2: Nano-Se (25) | 126 ± 1.00 f | 11.62 ± 0.07 c | 184.43 ± 0.51 d | 240 ± 1.52 c | 15.6 ± 0.40 d | 27.53 ± 0.56 ab | 7.72 ± 0.10 c | 0.0023 ± 0.0001 d |
| T3: Nano-Se (50) | 154 ± 1.00 e | 10.89 ± 0.10 e | 230.40 ± 0.52 b | 265 ± 1.00 a | 19.50 ± 0.50 b | 23.50 ± 0.50 c | 4.95 ± 0.04 e | 0.0033 ± 0.0002 c |
| T4: Nano-Se (75) | 490 ± 1.52 b | 10.33 ± 0.15 f | 183.50 ± 0.50 d | 200 ± 2.00 e | 17.00 ± 0.50 c | 17.67 ± 0.65 d | 5.16 ± 0.14 e | 0.0036 ± 0.0001 b |
| T5: Nano-Se (100) | 518 ± 1.52 a | 11.07 ± 0.02 d | 236.86 ± 0.15 a | 197 ± 2.02 e | 20.73 ± 0.75 a | 19.03 ± 0.47 d | 4.35 ± 0.14 f | 0.0057 ± 0.0001 a |
| T6: Nano-Cu (50) | 252 ± 1.00 c | 15.75 ± 0.05 a | 197.90 ± 0.40 c | 266 ± 1.06 a | 20.10 ± 0.3 ab | 28.52 ± 0.45 a | 64.42 ± 0.51 b | 0.0016 ± 0.0001 e |
| T7: Nano-Cu (100) | 182 ± 1.00 d | 15.91 ± 0.10 a | 237.86 ± 0.15 a | 255 ± 2.00 b | 7.76 ± 0.25 e | 26.45 ± 0.50 b | 141.8 ± 0.20 a | 0.0031 ± 0.0001 c |
| LSD 0.05 | 2.05 | 0.16 | 2.06 | 2.99 | 0.85 | 1.57 | 0.43 | 0.001 |

Treatments: T1: Control; T2: 25 ppm nano-Se; T3: 50 ppm nano-Se; T4: 75 ppm nano-Se; T5: 100 ppm nano-Se; T6: 50 ppm nano-Cu; T7: 100 ppm nano-Cu. Means in a column followed by a common letter are not significantly different at the 5% level by DMRT; values are means ± standard deviation (SD) from three replicates.

**Table 4.** Effect of applied nano-Cu and nano-Se on bioavailability of content of some nutrients (mg kg$^{-1}$) in banana transplants.

| Doses | N | P | K | Fe | Mn | Zn | Cu | Se |
|---|---|---|---|---|---|---|---|---|
| T1: Control | 4621 ± 3.6 d | 273 ± 1.00 c | 1982 ± 1.00 d | 3900 ± 10.00 f | 78.0 ± 1.00 g | 240 ± 1.00 c | 112 ± 1.52 c | 0.206 ± 0.003 g |
| T2: Nano-Se (25) | 4620 ± 3.5 d | 168 ± 1.52 f | 1902 ± 2.00 f | 3235 ± 5.00 g | 139 ± 1.00 f | 249 ± 1.00 bc | 99 ± 1.00 d | 0.340 ± 0.001 e |
| T3: Nano-Se (50) | 4901 ± 17 b | 210 ± 1.00 d | 2180 ± 2.00 c | 5425 ± 5.00 e | 206 ± 1.00 d | 275 ± 1.00 bc | 98 ± 1.00 de | 0.509 ± 0.002 b |
| T4: Nano-Se (75) | 5320 ± 2.0 a | 189 ± 1.00 c | 1919 ± 1.00 e | 8025 ± 5.00 a | 200 ± 1.00 e | 265 ± 1.52 bc | 96 ± 1.00 e | 1.610 ± 0.002 a |
| T5: Nano-Se (100) | 4760 ± 3.5 c | 168 ± 1.52 f | 2265 ± 1.00 b | 5925 ± 5.00 d | 210 ± 1.00 c | 264 ± 1.00 bc | 97 ± 1.00 de | 0.456 ± 0.012 c |
| T6: Nano-Cu (50) | 4901 ± 8.0 b | 279 ± 6.02 b | 1743 ± 1.00 g | 6075 ± 5.00 c | 387 ± 1.00 b | 285 ± 1.00 b | 128 ± 2.00 b | 0.256 ± 0.001 f |
| T7: Nano-Cu (100) | 4060 ± 5.0 e | 289 ± 5.56 a | 2318 ± 1.00 a | 6325 ± 45.00 b | 434 ± 1.00 a | 328 ± 58.60 a | 160 ± 2.00 a | 0.406 ± 0.001 d |
| LSD 0.05 | 13.88 | 5.73 | 12.10 | 10.46 | 1.75 | 38.83 | 2.50 | 0.008 |

Treatments: T1: Control; T2: 25 ppm nano-Se; T3: 50 ppm nano-Se; T4: 75 ppm nano-Se; T5: 100 ppm nano-Se; T6: 50 ppm nano-Cu; T7: 100 ppm nano-Cu. Means in a column followed by a common letter are not significantly different at the 5% level by DMRT; values are means ± standard deviation (SD) from three replicates.

### 3.5. Chemical Composition of Acclimatized Plants

The impact of applied different doses of nano-Se and nano-Cu on the chemical composition of banana transplants during acclimatization is presented in Table 4. Many important findings could be extracted from this Table such as (1) an increased transplant content of the studied nutrients (N, K, Fe, Mn, Se and Zn) by increasing applied nano-Se up to 100 mg L$^{-1}$, except for Cu and P, which recorded the opposite trend; (2) the highest values of all studied nutrients were significantly achieved by application of 100 mg L$^{-1}$ nano-Cu compared to nano-Se or control, except N; and (3) the highest values of all studied nutrients

were recorded with the application of an 100 mg $L^{-1}$ dose of nano-Cu compared to other treatments, except N, Fe, and Se.

## 4. Discussion

### 4.1. Growth of Acclimatized Plants

Micropropagation of banana is a successful protocol to produce healthy transplants or explants, which could be placed in artificial culture media in an aseptic and controlled environment [17]. During the current study, banana plantlets had the highest survivorship by 95.3% by application of the highest dose of nano-Cu (100 mg $L^{-1}$), whereas the highest dose of nano-Se recorded 94.3% (Table 2). The acclimatization of micropropagated banana plantlets was significantly observed through improving all studied vegetative growth parameters, shown in Tables 2 and 3 (i.e., number or roots and leaves, transplant height and its diameter) due to the soil application of nano fertilizers of Se and Cu. This improvement of transplants may be resulted from the effective role of nanofertilizers in enhancing the growth of banana transplants during acclimatization. A strong relationship between growth of transplants and nanofertilizers has been reported in the literature, but no published articles concern the soil application of Cu- and/or Se-nanofertilizers on the acclimatization of banana transplants. However, some published articles could be found on banana transplants on different issues like increasing the regeneration of plantlets by using nano-ZnO [34], mitigation stress of water deficit or salinity by applied Si-NPs [35,36], and improvement rooting by myco synthesized CuO-nanoparticles [37].

### 4.2. Photosynthetic Pigments and Fluorescence

The photosynthetic pigments (Chl a, Chl b, and carotenoids) of banana transplants during the acclimatization stage are crucial biological components because their efficiency may guarantee high survivorship of banana transplants. These components can support banana transplants with enough photosynthetic products, which strengthen the vegetative growth of transplants [38]. The applied nanofertilizers of Cu or Se increased the banana leaves' content of photosynthetic pigments up to 100 mg $L^{-1}$ of nano-Cu or nano-Se. This finding may answer the main question in this study: is using nanofertilizers effective in acclimatization and rooting of banana transplants? A strong relationship between plant photosynthetic pigments and nanofertilizers has been reported in the literature [39–46]. Surprisingly, nano-Cu recorded the highest values of photosynthetic pigments and its fluorescence compared to control or nano-Se. This finding may be linked to the role of copper in transporting the photosynthetic electron. Cu ions, as well, suppress the activity of the photosynthetic water splitting system. Copper effects on the photochemistry of photosystem II may reflect in decreasing $CO_2$ uptake in plant cells [47,48].

### 4.3. Antioxidant Enzymatic Activities of Banana Transplants

Due to the non-toxic applied doses of both nano-Cu and nano-Se fertilizers (100 mg $L^{-1}$), an increase in all antioxidant enzymatic activities (i.e., CAT, PPO and POX) were observed by increasing the applied nanofertilizers. These findings are not surprising given the fact that other research shows applied nano-Cu or Se did not cause any damage to the antioxidant system in many crops when they applied up to 100 mg $L^{-1}$ of, e.g., *Chrysanthemum morifolium* Ramat [40] and *Oryza sativa* [49]. Furthermore, high photosynthetic rates and enzyme activities in the banana transplants, as well as a high growth rate, were observed due to the high applied dose of nanofertilizers. The observed increase in banana enzymatic activities could be also attributed to the presence of Se or Cu, which are main components or cofactors in many plant enzymes like catalase, peroxidase, and polyphenol oxidase.

### 4.4. Growing Media and Acclimatized Plants

The growing medium supported the growth of banana transplants by preserving suitable nutrients and a low pH, as shown in Table 1. The applied nanofertilizers significantly improved the rooting and acclimatization of banana transplants as shown in the vegetative

growth, higher photosynthetic activity, enzymatic activities, and higher uptake of nutrients by cultivated transplants. There are several possible explanations for this result, such as applied nano-Cu having decreased the pH of the growing medium, which increased the uptake of micronutrients like Cu, Mn, and Zn, and this low pH may enhance the activity of the catalase enzyme. The comparison of nano-Se and nano-Cu gives priority to nano-Cu, which recorded higher values in general in all studied properties. The reason for this is not clear but it may have to do with the biological impact of nano-Cu in banana transplants and the physiological role. Few published articles on the impact of growing media and acclimatized banana plants could be noticed in the presence of applied nano-fertilizers. This new window needs to be opened to focus on the promising role of nanonutrients on acclimatized banana plants. In general, some studies discussed the role of growing media during the acclimatization and performance of banana using different substrates, such as in [50].

**5. Conclusions**

This study set out with the aim of assessing the importance of Cu and Se nanofertilizers in improving the rooting and acclimatization of banana transplants. This improvement could be noticed from enhancing the rooting of transplants (higher root length and number of roots), and promoting acclimatization by increasing the dry weight, diameter of transplants, and their survival rates. The photosynthetic activity in banana transplants, as well as the antioxidant enzymatic activities, were also improved by the application of nanofertilizers comparing to control. Both biochemical markers reached the highest values at the highest applied doses of nano-Se and nano-Cu. The bioavailability of the studied nutrients (N, P, K, Cu, Fe, Mn, Se, and Zn) to banana transplants were enhanced by the applied nanofertilizers, which may be resulted from the decrease of the pH and salinity of the growing medium. The applied doses of Se and Cu nanofertilizers (up to 100 mg $L^{-1}$) achieved the optimum growth conditions in this first report concerning the application of Se and Cu nanofertilizers for the acclimatization of banana transplants.

**Author Contributions:** Conceptualization and visualization, T.A.S., S.M.E.-B., M.E.E.-M., A.E.-D.O., H.S.E.-B. and H.E.-R.; resources, T.A.S., S.M.E.-B., M.E.E.-M., A.E.-D.O., H.S.E.-B. and H.E.-R.; methodology, T.A.S., S.M.E.-B., M.E.E.-M., A.E.-D.O., H.S.E.-B. and H.E.-R.; software, T.A.S., S.M.E.-B., M.E.E.-M., A.E.-D.O., H.S.E.-B. and H.E.-R.; validation, T.A.S., S.M.E.-B., M.E.E.-M., A.E.-D.O., H.S.E.-B. and H.E.-R.; investigation, T.A.S., S.M.E.-B., M.E.E.-M., A.E.-D.O., H.S.E.-B. and H.E.-R.; data curation, T.A.S.; writing—original draft preparation—review and editing, T.A.S., S.M.E.-B., M.E.E.-M., A.E.-D.O., H.S.E.-B. and H.E.-R.; funding acquisition, T.A.S., S.M.E.-B., M.E.E.-M., A.E.-D.O., H.S.E.-B. and H.E.-R. All authors have read and agreed to the published version of the manuscript.

**Funding:** Deanship of Scientific Research, Vice Presidency for Graduate Studies and Scientific Research, King Faisal University, Saudi Arabia (Grant No. NA000171).

**Institutional Review Board Statement:** Not applicable.

**Informed Consent Statement:** Not applicable.

**Data Availability Statement:** Not applicable.

**Acknowledgments:** Authors acknowledge the Deanship of Scientific Research Vice Presidency for Graduate Studies and Scientific Research, at King Faisal University, for the financial support, under Nasher Track (grant no.NA000171).

**Conflicts of Interest:** The authors declare no conflict of interest.

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
