# Peer review of "Acclimatization of In Vitro Banana Seedlings Using Root-Applied Bio-Nanofertilizer of Copper and Selenium"

_agronomy, doi:10.3390/agronomy12020539_

Round 1

Reviewer 1 Report

Dear Authors,

Your choice of preparing manuscript concerning optimal fertilization of one of the leading fruit crops worldwide is very significant. Banana is very important fruit as far from agricultural as well from biological point of view. Solving of the issue of effective application of nano-fertilizers in production of banana transplants has practical and scientific implications. Manuscript is well constructed and used methods enabled to justify the obtained results. Anyway I found some drawbacks – the first one is significance level you have applied i.e. P 0.05 but in Reviewer’s opinion this level can be used only in analysing of field trial. My recommendation is using of significance level 0.01. Generally, all figures in the manuscript are very well composed and prepared with the only exemption is Fig. 1 (survival rate). In Reviewer’s opinion Y axis should always start at value of zero. Also Reviewer would like to suggest instead of T1 for the control treatment using symbol “0” or C (control). Reviewer as non-native English speaker is not able to evaluate English language in details. Anyway in many places meaning of the sentences is obscure because of grammar and style problems There are some examples below: 

lines 25-26 – unclear sentence;

line 43- “foods” ;

46-47 – unclear sentence;

Line 49 – lack of square bracket;

Reviewer would like to suggest transferring lines 52-54 to the next paragraph (after words “wheat [7]”.

Line 58 – rather carbohydrates not carbohydrate;

Line 60 – unclear sentence as I can guess it should stand “may be controlled by”;

Lines 62-64 – problems in style;

Lines 192-193 – instead of “variable fluorescence” Reviewer would suggest “range of fluorescence”;

Lines 209-212; 254-260; 269 -270 – serious problems with grammar.

Therefore English language editing is absolutely necessary.

On the other hand Reviewer would like to congratulate Authors for preparing Reference section where Reviewer could not find any mistake

Author Response

Reviewer 1#

Comments and Suggestions for Authors

lines 25-26 – unclear sentence;

Response: done, thanks!

line 43- “foods” ;

Response: done, thanks!

46-47 – unclear sentence;

Response: done, thanks!

Line 49 – lack of square bracket;

Response: done, thanks!

Reviewer would like to suggest transferring lines 52-54 to the next paragraph (after words “wheat [7]”.

Response: done, thanks!

Line 58 – rather carbohydrates not carbohydrate;

Response: done, thanks!

Line 60 – unclear sentence as I can guess it should stand “may be controlled by”;

Response: done, thanks!

Lines 62-64 – problems in style;

Response: done, thanks!

Lines 192-193 – instead of “variable fluorescence” Reviewer would suggest “range of fluorescence”;

Response: done, thanks!

Lines 209-212; 254-260; 269 -270 – serious problems with grammar.

Response: done, thanks!

Therefore English language editing is absolutely necessary.

Response: thanks!

On the other hand Reviewer would like to congratulate Authors for preparing Reference section where Reviewer could not find any mistake

Response: thanks!

Reviewer 2 Report

The manuscript includes very interesting results. The introduction provides sufficient background and includes the latest relevant references. The results are quite clearly presented and discussed. However, there are a few things that need to be corrected or explained:

2. Materials and Methods: The number of replications is not included in this section for each type of measurement.

Figure 3: How was the control in the left figure containing plants for Nano-Cu different from the control in the right figure containing plants for Nano-Se? These control plants look a little different.

line 185: "3.1. Subsection" should be deleted.

line 252: What does the phrase 'environment 17' mean?

4.4. Growing media and acclimatized plants: Why did the authors not discuss this in relation to the literature data? No references are cited.

Author Response

Reviewer 2#

The manuscript includes very interesting results. The introduction provides sufficient background and includes the latest relevant references. The results are quite clearly presented and discussed. However, there are a few things that need to be corrected or explained:

  1. Materials and Methods: The number of replications is not included in this section for each type of measurement.

Response: thanks! All replicated numbers were added in ln 82, 11, 121, 129 and 145

Figure 3: How was the control in the left figure containing plants for Nano-Cu different from the control in the right figure containing plants for Nano-Se? These control plants look a little different.
Response: thanks!

This is just different plantlet. We did not use the same plantlet in the photos, thanks!

line 185: "3.1. Subsection" should be deleted.

Response: deleted, thanks!

line 252: What does the phrase 'environment 17' mean?
Response: corrected, thanks! The number is ref, number and the [ ] was added, thanks!

4.4. Growing media and acclimatized plants: Why did the authors not discuss this in relation to the literature data? No references are cited.
Response: added, thanks!

Reviewer 3 Report

I have completed the review of the article: Acclimatization of Banana Transplants in Vitro Using a Copper and Selenium Root-Applied Bio-Nano-Fertilizer.
It is an interesting and topical study in the context of the importance of the propagation of banana crops worldwide. The strong point are the treatments used and the experimental design in the context of the need to optimize the efficiency of fertilizers. Specific comments:
1. L35, please be careful when writing words, you wrote decease instead of decrease.
2. L49, please put [] to citation 4.
3. L86-89 please reformulate this sentence. Also, please briefly present the results obtained by this characterization technique in the methods section or results, where you consider appropriate.
4. For figures, please specify what the lowercase letters represent. Please specify what T1..T7 represents in methods (L91-102) so that the description of the figures is as concise as possible and therefore without this specification.
5. Please pay attention to the tables, 1 and 2 have the description of the treatments and 3 and 4 have T1..T7, please standardize the text to be the same everywhere.

Author Response

Reviewer 3#

I have completed the review of the article: Acclimatization of Banana Transplants in Vitro Using a Copper and Selenium Root-Applied Bio-Nano-Fertilizer.
It is an interesting and topical study in the context of the importance of the propagation of banana crops worldwide. The strong point are the treatments used and the experimental design in the context of the need to optimize the efficiency of fertilizers. Specific comments:

  1. L35, please be careful when writing words, you wrote decease instead of decrease.

Response: corrected thanks!

  1. L49, please put [] to citation 4.

Response: added, thanks!

  1. L86-89 please reformulate this sentence. Also, please briefly present the results obtained by this characterization technique in the methods section or results, where you consider appropriate.

Response: added in M and M section, thanks!

  1. For figures, please specify what the lowercase letters represent. Please specify what T1..T7 represents in methods (L91-102) so that the description of the figures is as concise as possible and therefore without this specification.

Response: it is ok, thanks!       

  1. Please pay attention to the tables, 1 and 2 have the description of the treatments and 3 and 4 have T1..T7, please standardize the text to be the same everywhere.

Response: Done, thanks!
